# Clinical Significance of Tumor Size, Pathological Invasion Sites Including Urinary Collecting System and Clinically Detected Renal Vein Thrombus as Predictors for Recurrence in pT3a Localized Renal Cell Carcinoma

**DOI:** 10.3390/diagnostics10030154

**Published:** 2020-03-12

**Authors:** Takuto Shimizu, Makito Miyake, Shunta Hori, Kota Iida, Kazuki Ichikawa, Keiichi Sakamoto, Tatsuki Miyamoto, Yasushi Nakai, Takeshi Inoue, Satoshi Anai, Nobumichi Tanaka, Kiyohide Fujimoto

**Affiliations:** Department of Urology, Nara Medical University, 840 Shijo-cho, Kashihara, Nara 634-8522, Japan; takutea19@gmail.com (T.S.); makitomiyake@yahoo.co.jp (M.M.); horimaus@gmail.com (S.H.); kota1006ida@yahoo.co.jp (K.I.); aburatani40@gmail.com (K.I.); keitsubaki2001@yahoo.co.jp (K.S.); tatsuki8770@gmail.com (T.M.); nakaiyasushi@naramed-u.ac.jp (Y.N.); inoue620@naramed-u.ac.jp (T.I.); sanai@naramed-u.ac.jp (S.A.); sendo@naramed-u.ac.jp (N.T.)

**Keywords:** renal cell carcinoma, recurrence, T3a, tumor size, urinary collecting system, renal vein thrombus

## Abstract

The recent eighth tumor-node-metastasis (TMN) staging system classifies renal cell carcinoma (RCC) with perirenal fat invasion (PFI), renal sinus fat invasion (SFI), or renal vein invasion (RVI) as stage pT3a. However, limited data are available on whether these sites have similar prognostic value or recurrence rate. We investigated the recurrence rate based on tumor size, pathological invasion sites including urinary collecting system invasion (UCSI), and clinically detected renal vein thrombus (cd-RVT) with pT3aN0M0 RCC. We retrospectively reviewed 91 patients with pT3aN0M0 RCC who underwent surgical treatment. Patients with tumor size > 7 cm, UCSI, three invasive sites (PFI + SFI + RVI), and cd-RVT showed a significant correlation with high recurrence rates (hazard ration (HR) 2.98, *p* = 0.013; HR 8.86, *p* < 0.0001; HR 14.28, *p* = 0.0008; and HR 4.08, *p* = 0.0074, respectively). In the multivariate analysis, tumor size of >7 cm, the presence of UCSI, and cd-RVT were the independent predictors of recurrence (HR 3.39, *p* = 0.043, HR 7.31, *p* = 0.01, HR 5.06, *p* = 0.018, respectively). In pT3a RCC, tumor size (7 cm cut-off), UCSI, and cd-RVT may help to provide an early diagnosis of recurrence.

## 1. Introduction

Renal cell carcinoma (RCC) accounts for approximately 3% of all reported human cancers worldwide [1]. The gold standard for the treatment of localized RCCs is radical nephrectomy (RN) or partial nephrectomy (PN). Despite complete surgical resection, >25% of patients progress to metastatic disease [2]. Once metastasized, the five-year survival rate of RCC is <10% [3]. Therefore, it is important to classify patients with RCC according to recurrence risk and to identify patients who need short-term follow-up and pre- or postoperative treatment with molecularly targeted drugs or immune checkpoint inhibitors. 

The recent eighth edition of the Union for International Cancer Control (UICC) tumor-node-metastasis (TNM; 2017) classification system is currently used for staging RCC; pT stage is defined by tumor size and local tumor extension. Of these, pT3a RCC is pathologically heterogeneous because the eighth edition of the TNM system classifies tumors with perirenal fat invasion (PFI), renal sinus fat invasion (SFI), or renal vein invasion (RVI) as stage pT3a [4]. Limited data are available on whether the sites of invasion have similar prognostic value or recurrence rate, and these data are controversial [5,6,7,8,9]. Bedke et al. reported that PFI and SFI simultaneously and independently influenced cancer-specific mortality (CSM) [5]. In contrast, Poon et al. found that there was no difference between PFI and RFI in terms of prognostic impact [6]. Costa et al. and Baccos et al. found an increase in CSM in patients with simultaneous PFI and RVI [7,8].

There are also reports that cT3a needs to be divided by 7 cm as well as distinguishing cT1 and cT2 instead of invasive sites [9]. Brookman-May et al. reported that the prognostic impact of PFI and RVI on CSS appeared to be comparable. They suggested that applying a 7 cm tumor size cut-off within another staging system may allow for enhanced prognostic discrimination of pT3a RCC [9]. Furthermore, there are controversial studies on the prognostic role of urinary collecting system invasion (UCSI) in RCC [10,11]. Waalkes et al. reported that UCSI is not an independent prognostic factor for RCC [10]. In contrast, Chen et al. argued that UCSI has a significant negative impact on overall survival and recurrence-free survival in RCC patients and could be used to predict cancer-specific survival especially in localized RCC. Thus, RCC patients with UCSI should be paid more attention by clinicians and pathologists and require close follow-up for their poor prognosis [11]. In addition, it is unclear whether the treatment of renal vein thrombus detected by preoperative image modality should be similar to that detected by postoperative pathological findings. In this context, Ball et al. found that segmental RVI was associated with better RFS and CSS than main RVI [12].

Therefore, we performed a comprehensive analysis to identify the predictors of recurrence after RN or PN in patients with pT3a localized RCC.

## 2. Materials and Methods

### 2.1. Patients Selection

We retrospectively analyzed the data of 424 patients who underwent unilateral RN or PN for RCC at our institute between 2007 and 2017. There were 91 patients with pT3aN0M0 RCC according to the 2017 TNM staging system (Figure 1). A histopathological review was conducted by an experienced uropathologist to determine the pathological T category, Fuhrman grade, infiltration (INF), the presence of necrosis, sarcomatoid variant, histological classification, lymphovascular invasion (LVI), and UCSI. Furthermore, medical imaging findings were reviewed by an experienced radiologist specializing in the urogenital system to determine the clinical T category.

Histopathologically confirmed (pN0) and clinically uninvolved lymph nodes (cN0) were defined as stage N0. All tumors were restaged according to the 2017 TNM system. The radiologist used abdominal computed tomography (CT) or magnetic resonance imaging (MRI) and chest imaging (CT or chest X-ray) for clinical staging of RCC. None of the patients received pre- or postoperative therapy. The follow-up care after surgery required chest and abdominal imaging studies (abdominal ultrasound and thoracoabdominal CT) 2–4 times a year. Recurrence was defined as distant metastasis and local recurrence identified by CT scan.

### 2.2. Ethical Approval

This study was approved by the institutional review board (IRB) of the Nara Medical University (Nara, Japan; Medical Ethics Committee ID: NMU-1966; approval date 10 October 2019) and complied with the 1964 Helsinki Declaration and its later amendments. As data for the study were obtained through a retrospective review, a waiver of informed consent was approved by the IRB. Personal information of the subjects and donors was anonymized when necessary; the information was labeled with an identifying code to make it possible to distinguish between the individuals. Then, deidentified patient data were analyzed.

### 2.3. Statistical Analysis

For statistical analysis, the survival intervals were calculated from the date of surgery to the date of first diagnosis of metastasis (recurrence-free survival (RFS)), the date of death from cancer (CSS), or the last follow-up. Statistical analyses were performed using SPSS for Windows (version 20.0; IBM, Corp., Armonk, NY, USA). Figures were generated using GraphPad Prism 5.0 (GraphPad Software, Inc., La Jolla, CA, USA). RFS and CSS were estimated using the Kaplan–Meier method and the log-rank test was applied to compare survival curves. The univariate and multivariate Cox proportional-hazards regression models were used to identify the predictors of recurrence. A *p*-value < 0.05 was considered statistically significant.

## 3. Results

### 3.1. Patient Characteristics

We identified 91 patients with pT3aN0M0 RCC of the 424 RCC patients who underwent surgery at our institute from 2007 to 2017. The clinical and pathological characteristics of the 91 patients are summarized in Table 1.

There were 65 men (71.4%) and 26 women (28.6%). The median age was 68 years old (range, 41–87 years). The median follow-up period was 48 months (range, 6–134 months). During the follow-up period, 26 patients (28.6%) experienced recurrence, and 11 patients (12.1%) died; of these, 10 (11.0%) died due to recurrence and progression of RCC. Overall, the actuarial five-year RFS rate was 65.3% and the five-year CSS rate was 88.4% (Figure 2A,B).

There were 35 patients (38.5%) with cT1 and 16 patients (17.6%) with cT2 RCC. Seven patients (7.6%) underwent PN. Clinically detected renal vein thrombus (cd-RVT) was observed in 17 patients (18.7%) but not in the remaining 74 patients (81.3%). The median tumor size was 6.4 cm (range, 1.4–14.0 cm). The tumor size was ≤7 cm in 62 patients (68.1%) and was >7 cm in 29 patients (31.9%). UCSI was observed in 22 patients (24.2%) but not in the remaining 69 patients (75.8%). Forty-six patients (50.5%) had RVI. PFI was observed in 34 patients (37.3%); 55 patients (60.4%) had SFI. There were 56 patients (61.5%) with infiltration in only one site, 26 (28.5%) with infiltration in two sites, and 9 (9.9%) with infiltration in three sites.

### 3.2. Exploration of Prognostic Factors for RFS and CSS

We studied the impact of tumor size, UCSI, invasion site, and renal vein thrombosis on RFS. First, we found a statistically significant difference between patients with tumor size ≤ 7 cm and those with tumor size > 7 cm, and the five-year RFS rate was 71.9% and 50.9%, respectively (hazard ratio (HR) 2.98, *p* = 0.013, Figure 3a). Second, there was a statistically significant difference between patients with and without UCSI, and the five-year RFS rate was 17.3% and 76.7%, respectively (HR 8.86, *p* < 0.0001, Figure 3b). Third, there was a statistically significant difference between patients with one or two invasive sites and those with three invasive sites (PFI + SFI + RVI), and the five-year RFS rate was 69.5% and 27.8%, respectively (HR 14.28, *p* = 0.0008, Figure 3c). Finally, we found that there was a statistically significant difference between patients with and without cd-RVT, and the five-year RFS rate was 24.9% and 74.7%, respectively (HR 4.07, *p* = 0.0074, Figure 3d). Further, the Cox univariate analyses showed that a high C-reactive protein (CRP), high Fuhrman grade, INF b or c, the presence of LVI, necrosis, and RVI were predictive factors for recurrence and poor prognosis. In contrast, in the univariate analysis, gender, age, histological classification, sarcomatoid variant, PFI, and SFI showed no significant difference in predicting RFS (Table 2).

In multivariate analysis for recurrence, tumor size > 7 cm (HR 3.27, *p* = 0.004), the presence of UCSI (HR 4.26, *p* = 0.001), and cd-RVT (HR 4.10, *p* < 0.001) were found to be independent prognostic factors for predicting RFS (Table 2). We limited the number of variables to three considering the potential to overfit the model due to too many variables for the number of events (26 events) in a multivariable model.

Similarly, for CSS, the univariate analysis showed that there was a statistically significant difference between patients with one or two invasive sites and those with three invasive sites, and the five-year CSS rate was 93.8% and 44.4%, respectively (HR NA, *p* < 0.0001, Figure 4c), as well as between patients with and without cd-RVT, and the five-year CSS rate was 70.5% and 92.6%, respectively (HR 5.08, *p* = 0.049, Figure 4d). However, there was no statistically significant difference in CSS between patients in terms of tumor size (HR 3.25, *p* = 0.07, Figure 4a) and the presence of USCI (HR 4.35, *p* = 0.07, Figure 4b). However, patients with tumor size > 7 cm or with USCI showed a tendency to have poor CSS. This could be due to the small sample size or the short observation period. In this study, of the 91 patients, only 10 patients died due to RCC. In the univariate analysis, gender, age, CRP, Fuhrman grade, INF, LVI, necrosis, histological classification, sarcomatoid variant, PFI, and SFI showed no significant difference in predicting CSS (Table 3). We do not show the results of multivariate analysis for CSS because there are few cases of cancer death.

In addition, we performed a review on UCSI and hematuria based on preoperative urinalysis, and found a correlation between the presence of UCSI and the presence of hematuria (Table 4).

## 4. Discussion

In this study, we found that tumor size of >7 cm, the presence of USCI, and cd-RVT were independent predictors for the poor prognosis of metastasis or recurrence in pT3aN0M0 RCC after surgery.

According to the latest TNM classification, although T1 and T2 are classified by size, T3 is classified by the site of tumor invasion [4]. However, several recent studies have reported that a 7 cm tumor size cut-off is useful for predicting the prognosis of pT3a RCC [9,12]. We examined the impact of tumor size on the risk of RFS and CSS. In this study, 91 patients with pT3aN0M0 RCC were divided into two groups; 62 patients with tumor size ≤ 7 cm and 29 patients with tumor size > 7 cm. The five-year CSS rate for patients with a tumor size ≤ 7 cm and >7 cm was 92.1% and 80.2%, respectively (HR 3.25, *p* = 0.07, Figure 4a). There was no statistically significant difference in CSS between the two groups; however, the patients with tumor size > 7 cm showed a tendency to have poor CSS. The lack of statistical significance difference may be due to the small sample size or the short observation period. The five-year RFS rate was 71.9% and 50.9% for patients with tumor size ≤ 7 cm and >7 cm, respectively (HR 2.98, *p* = 0.0125, Figure 3a). Furthermore, the multivariate Cox proportional-hazard analysis showed that tumor size > 7 cm was independently associated with an increased risk of recurrence (HR 3.27; *p* = 0.004). This result is consistent with the findings of Chen et al. They analyzed 163 patients with pT3aN0M0 RCC and demonstrated that patients with tumor size > 7 cm was associated with worse estimated five-year RFS or CSS, compared with those with tumor size ≤ 7 cm [13]. In addition, they identified that tumor size was an independent predictor for RFS and CSS by the multivariate Cox proportional-hazard analysis [13]. In the report, the size of the tumor is related to the recurrence rate: the larger is the tumor, the more likely it is to be high-grade. Conversely, those with high malignancy have high tumor growth ability and may be found in an enlarged state. In the study, compared with <7-cm tumor, ≥7-cm tumor significantly more frequently had Fuhrman grade 3–4 (29.7% vs. 51.6%; *p* = 0.009) [13]. Brookman-may et al. reported that, compared with stage pT1–2, patients with pT3a RCC significantly more frequently had Fuhrman grade 3–4 (29.4% vs. 13.4%; *p* < 0.001) and had tumor with >7 cm size (13% vs. 39.1%; *p* < 0.001). From this, it is expected that the malignancy will worsen as the size increases, but there was no description of the actual comparison between size and Fuhrman grade [9]. In fact, our study also showed a significant correlation between tumor size and Fuhrman grade/INF/growth pattern in the spearman r test (Appendix A).

In this study, we examined the pathological invasion sites (PFI, SFI and RVI). Of the three sites, only RVI showed significant differences in recurrence rates in the univariate analysis (HR 2.42, *p* = 0.022); however, no significant difference was found in the multivariate analysis (HR 1.73, *p* = 0.39). In addition, the presence or absence of any pathological invasion site was not significantly different in CSS. In contrast, the tumors invading all three sites showed significant difference in RFS (HR 14.3, *p* = 0.0008) and CSS (HR NA, *p* < 0.0001) in the univariate analysis, but not in the multivariate analysis. 

RCC often have a pseudocapsule. It can be hypothesized that pseudocapsules may have a protective role against tumor invasion to perirenal fat. In contrast, pseudocapsules have a loss of continuity on the hilum side, I mean on the side of the renal sinus fat, and it is rich in adipose tissue and has numerous veins and lymph vessels in renal sinus. Therefore, it is theoretically thought that SFI may increase the recurrence rate, but in this study there was no significant difference between the two fat sites (SFI and PFI), as in previous report [5,6,9]. Exceeding the pseudocapsule itself may increase the degree of malignancy. Minervini et al. reported that, compared with tumors free from invasion to pseudocapsule, tumors with invasion to pseudocapsule had significantly association with pathologic tumor dimensions and grade [14]. In this study, RVI performed worse than PFI or SFI. Certainly, it is not difficult to predict that the recurrence rate seems to be higher with invasion into blood vessels than with direct invasion into fats and that the prognosis is very poor if it has all three sites invasion, as shown in this study. However, the current TNM classification does not mention the number of invasion sites. The results of this study are consistent with previous reports. Costa et al. analyzed 46 patients with pT3aNxMx RCC and Baccos et al. analyzed 122 patients with pT3aNxMx RCC. They found that an increase in CSM in patients with simultaneous PFI and RVI [7,8]. Furthermore, Brookman-May et al. reported the prognostic impact of PFI and RVI on CSM seems to be comparable [9]. In any case, evaluation of these three invasive sites can play an important role in predicting the patient’s prognosis and considering adjuvant therapy.

It is unclear whether clinically and pathologically detected renal vein thrombus should be treated the same as pT3a. Therefore, we detected renal vein thrombus not only pathologically but also through diagnostic imaging studies. As a result of cd-RVT, a tumor thrombus that remains in the renal vein and does not extend to the inferior vena cava showed significant difference in RFS (HR 4.07, *p* = 0.0074) and CSS (HR 5.08, *p* = 0.049) in the univariate analysis. Furthermore, in the multivariate analysis, cd-RVT was an independent prognostic factor for RFS (HR 4.10, *p* < 0.001). This result is consistent with the study conducted by Utsunomiya et al. in Japan [15]. They also pointed out, based on the results of the univariate analysis, that renal vein tumor thrombus, which appears clear in diagnostic imaging studies, is a predictive factor for postoperative recurrence of pT3aN0M0 RCC. Ball et al. investigated a similar question between T3a with segmental RVI (*n* = 87) and T3a with main RVI (*n* = 64) [12]. They reported that both RFS and CSS were significantly worse in patients with main RVI than those with segmental RVI. In the multivariate analysis, main RVI had inferior RFS and CSS compared with segmental RVI. Although segmental RVI or main RVI were determined by postoperative pathological examination, renal vein thrombus can be clinically detected preoperatively. Thus, paying attention to renal vein thrombus can lead to more adequate follow-up and use of preoperative treatment. It seems that there is no way to treat these the same among patients who underwent partial resection of cT1a tumors and those who underwent nephrectomy due to the presence of renal vein thrombus. PN used to be widely performed for the treatment of small RCC. It is unclear whether there is a difference between upstaged tumors and pT1 tumors. If the preoperative diagnosis is cT1a, even if the tumor is upstaged to pT3a, a relatively good outcome is possible. In contrast, if the preoperative diagnosis is cT1b, the risk of recurrence may increase if it is upstaged to pT3a [16,17]. Jeong et al. examined 896 patients with pT1 tumor and 91 patients with pT3a upstaged tumor; they reported that the two-year RFS was worse in the pT3a upstaged group than that in the pT1 group. Upstaging was associated with advanced age, cT1b stage, clinical symptoms, and a high Fuhrman grade in the multivariate analysis [16]. Lee et al. examined 1324 patients with pT1a tumors and 43 patients with pT3a upstaged tumor; they did not find any significant predictor for RFS, CSS, or overall survival [17].

In recent years, UCSI has been considered as a prognostic factor for RCC. Waalkes et al. assessed 1678 patients with RCC and reported that UCSI is not an independent prognostic factor for RCC [10]. In contrast, in pT3a RCC, Chen et al. examined 218 patients with RCC and argue that UCSI needs attention as an independent poor prognostic factor [11]. In our study, the presence of UCSI was significantly correlated with poor prognosis in the univariate analysis for RFS (HR 8.86, *p* < 0.0001) and CSS (HR 4.35, *p* = 0.07). In the multivariate analysis, it was identified as a prognostic factor for RFS (HR 4.26, *p* = 0.001). Further, we examined UCSI and hematuria via preoperative urinalysis. There is the correlation between the presence of UCSI and hematuria (Table 4). Considering the results of our study on the relationship between hematuria and UCSI, preoperative hematuria could be a marker for the presence of UCSI.

In our study, we found that tumor size > 7 cm, the presence of USCI, and cd-RVT were independent predictors for the poor prognosis for metastasis or recurrence in patients with pT3aN0M0 RCC after surgery. These findings are useful for reviewing pT3a staging and should be taken into consideration in future prospective studies. In addition, it can form the basis of a prospective study to determine whether pre- or postoperative treatment with molecular targeted drugs or immune checkpoint inhibitors should be used. At present, it is unclear whether RCC should be treated with these therapeutic agents before or after surgery. Patients with tumor size ≤ 7 cm, micro or macro hematuria findings, and cd-RVT have the potential to benefit from pre- or postoperative use of molecular-targeted drugs or immune checkpoint inhibitors. Furthermore, the main purpose of postoperative follow-up is to detect recurrence and metastasis at an early stage. Early detection of recurrence and metastasis enables resection; if resection is difficult, various other treatments (immunotherapy and molecular targeted therapy) may be effective. Although the American Urological Association (AUA), National Comprehensive Cancer Network (NCCN), and European Association of Urology (EAU) guidelines recommend postoperative follow-up according to the risk, there is not sufficient evidence [18,19,20]. The knowledge about predictors for recurrence can lead to adequate use of adjuvant therapy, early diagnosis of recurrence, and early intervention, resulting in improved prognosis of patients with pT3aN0M0 RCC. According to currently available TNM classification, pT3a RCC is pathologically heterogeneous. There are possibilities for improving the classification to predict prognosis more precisely.

Attention has been paid to the fact that circulating tumor cells (CTC) and cell-free tumor DNA (ctDNA) may also be useful in RCC for personalized therapy or selection of therapeutic drugs [21].

It is interesting to see whether RVI and cd-RVT increase CTC and ctDNA, and whether there is a change in urine markers with UCSI. Smith et al. reported that detection of ctDNA in plasma was more frequent in patients with larger tumors and in those with venous tumor thrombus. It is expected that metastasis is likely to occur if there are many CTC or ctDNA [22]. This report supports the results of our study. However, in our study, we did not pay attention to liquid biopsy. Further research is needed in CTC or ctDNA in RCC.

There were several limitations in this study. First, this study had a retrospective design, a relatively small sample size, and the follow-up period was short. Due to the small number of events (26 for RSS, 10 for CSS), we limited the number of variables for RSS considering the potential to overfit the model and did not show the multivariable analyses model for CSS. The data on the type of metastasis or treatment after relapse were not available for all patients. Hence, these factors were not analyzed. Besides, details regarding the molecular makers and nephrometry were not included in our present analysis. In contrast, laboratory data such as CRP and albumin and pathological factors such as LVI, necrosis, and sarcomatoid variant were studied. Furthermore, to the best of our knowledge, it is the first report simultaneously evaluating tumor size, USCI, and renal vein tumor thrombosis as predictors of RCC.

## 5. Conclusions

Our findings reveal the significant influence of tumor size (7 cm cut-off), UCSI, and cd-RVT for the poor prognosis of metastasis or recurrence in patients with pT3aN0M0 RCC after surgery. Thus, tumor size, UCSI, and cd-RVT may be considered for inclusion in the TNM classification of stage pT3a RCC in the future. Nevertheless, further research is required to validate these findings.

## Figures and Tables

**Figure 1 diagnostics-10-00154-f001:**
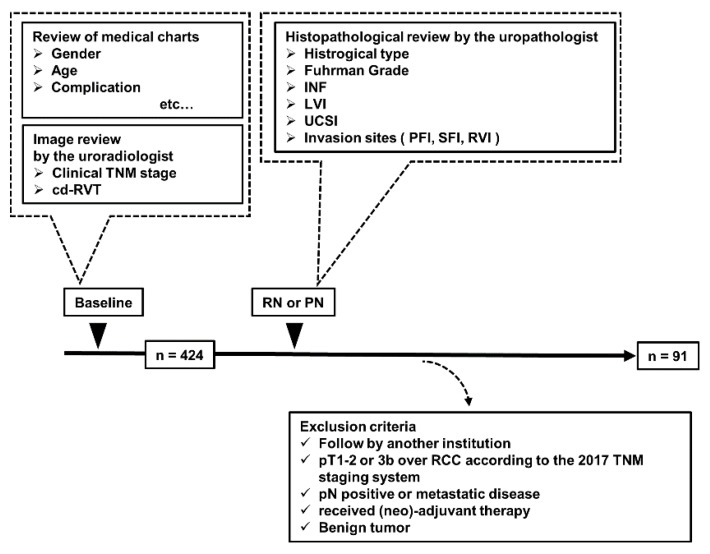
Schematic representation of the study workflow This study included 424 patients who underwent unilateral RN or PN for RCC at our institute between 2007 and 2017. Of these, 91 patients with pT3aN0M0 RCC were identified. Information regarding the preoperative conditions was obtained retrospectively by reviewing the medical records and laboratory blood tests. Data on gender, age, comorbidities, and C-reactive protein were obtained. Preoperative imaging evaluation and TNM classification including the presence or absence of cd-RVT were performed by a skilled uro-radiologist. The operation was performed by skilled urologists at our hospital. Postoperative sample evaluation was performed by a pathologist specializing in the urology. After establishing the exclusion criteria, 91 cases were identified. cd-RVT, clinically detected renal vein thrombus; INF, infiltration; LVI, lymphovascular invasion; UCSI, urinary collecting system invasion; RN, radical nephrectomy; PN, partial nephrectomy.

**Figure 2 diagnostics-10-00154-f002:**
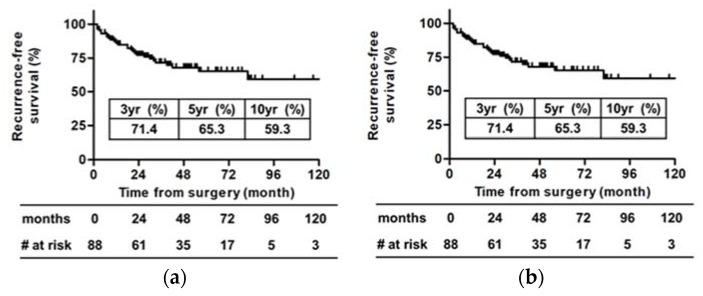
Recurrence-free survival and cancer-specific survival probabilities for pT3aN0M0 RCC. RFS and CSS were estimated using the Kaplan–Meier method. (**a**) The 3-, 5-, and 10-year RFS rates were 71.4%, 65.3%, and 59.3%, respectively. (**b**) The 3-, 5-, and 10-year CSS rates were 93.8%, 88.4%, and 81.1%, respectively RFS, recurrence-free survival; CSS, cancer-specific survival.

**Figure 3 diagnostics-10-00154-f003:**
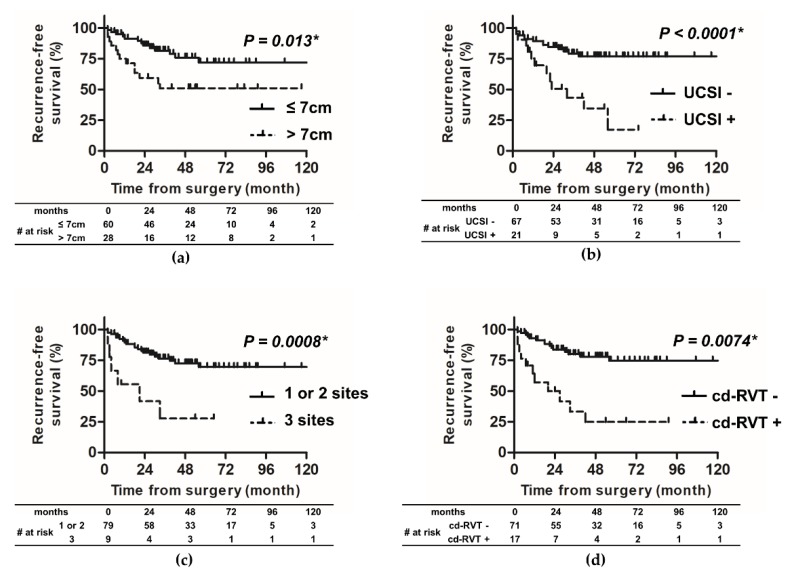
Recurrence-free survival probabilities for four main factors; tumor size, urinary collecting system invasion, pathological invasion sites, and clinically detected renal vein thrombus. Patients with tumor size > 7 cm (**a**); UCSI (**b**); three invasion sites (**c**); or cd-RVT (**d**) had a significantly higher risk of recurrence than those without. RFS, recurrence-free survival; UCSI, urinary collecting system invasion; cd-RVT, clinically detected renal vein thrombus.

**Figure 4 diagnostics-10-00154-f004:**
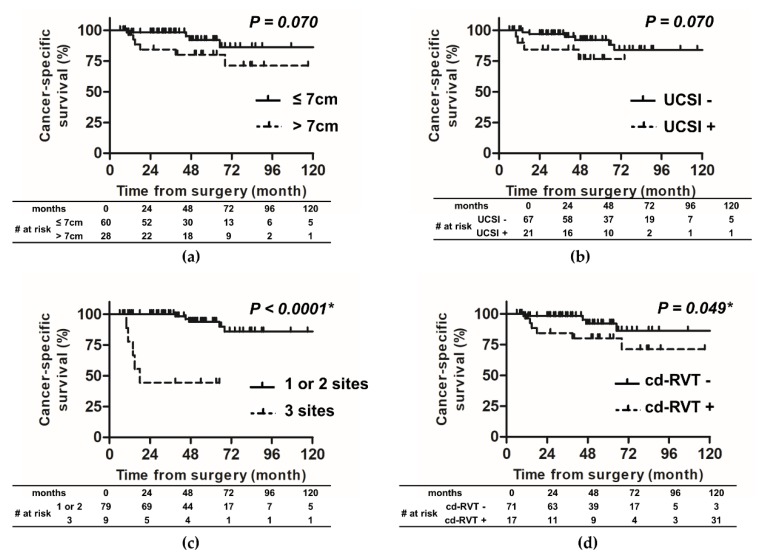
Cancer-specific survival probabilities for four main factors: tumor size, urinary collecting system invasion, pathological invasion sites, and clinically detected renal vein thrombus. Patients with three invasion sites (**c**) or cd-RVT (**d**) had a significantly higher risk than those without. On the other hand, there was no significant difference in CSS between patients with tumor size ≤ 7 cm or >7 cm (**a**). There was no statistically significant difference in CSS between patients with UCSI or without UCSI (**b**). CSS, cancer-specific survival; UCSI, urinary collecting system invasion; cd-RVT, clinically detected renal vein thrombus.

**Table 1 diagnostics-10-00154-t001:** Clinicopathological information.

Variables		Number of Patients	%
		(n = 91)	100
Gender	Male Female	65 26	71.4 28.6
Age	Median (Range)	68 (41–87)	
Follow-up (month)	Median (Range)	48 (6–134)	
Charlson comorbidity index	0 ≥1	41 50	45.1 54.9
CRP	<0.5 ≥0.5	61 30	67.0 33.0
Clinical T stage	cT1a/1b cT2a/2b cT3a/3b/3c cT4	14/21 7/9 32/6/0 2	15.4/23.1 7.7/9.9 35.2/6.6/0 2.2
Clinically detected renal vein thrombus	− +	74 17	81.3 18.7
Type of Surgery	RN PN	84 7	92.3 7.7
Tumor size (cm)	Median (Range)	6.4 (1.4–14)	
	≤7 >7	62 29	68.1 31.9
Histological type	Clear cell Papillary Chromophobe others	74 11 3 3	81.3 12.1 3.3 3.3
Fuhrman Grade	1–2 3–4 NA	39 39 13	42.9 42.9 14.2
INF	a b c	16 73 2	17.6 82.4 2.2
LVI	− +	29 61	31.9 67.0
Urinary collecting system invasion	− +	69 22	75.8 24.2
Pathologically invasion subgroups			
Renal vein invasion	− +	45 46	49.5 50.5
Peri renal fat invasion	− +	57 34	62.6 37.4
Renal sinus fat invasion	− +	36 55	39.6 60.4
Pathologically invasion number of subgroups	1 2 3	56 26 9	61.5 28.6 9.9

CRP, C-reactive protein; RN, radical nephrectomy; PN, partial nephrectomy; INF, infiltrative growth; LVI, lymphovascular invasion.

**Table 2 diagnostics-10-00154-t002:** Univariate and multivariate cox regression analyses for RFS (* statistically significant).

Variables		Univariate Analysis			Multivariate Analysis		
		HR	95% CI	*p* Value	HR	95% CI	*p* Value
Gender	Male	1		0.37			
	Female	1.48	0.61–3.57				
Age	≥70	1		0.26			
	<70	1.55	0.72–3.37				
CRP	<0.5	1		0.009 *	not included		
	≥0.5	3.05	1.32–7.06				
clinically detected renal vein thrombus	−	1		0.0074 *	1		<0.001 *
	+	4.07	1.45–11.37		4.10	1.86–9.01	
Fuhrman Grade	G1,2	1		0.015 *	not included		
	G3,4	2.97	1.23–7.17				
INF	a	1		0.011 *	not included		
	b/c	3.53	1.33–9.38				
LVI	−	1		0.03 *	not included		
	+	2.39	1.06–5.37				
Necrosis	−	1		0.0003 *	not included		
	+	4.26	1.96–9.28				
Histological classification	non-ccRCC	1		0.44			
	ccRCC	0.64	0.21–1.97				
Sarcomatoid variant	−	1		0.94			
	+	1.08	0.13–8.77				
Tumor size	≤7	1		0.013 *	1		0.004 *
	>7	2.98	1.26–7.05		3.27	1.46–7.31	
Urinary collecting system invasion	−	1		<0.0001 *	1		0.001 *
	+	8.86	3.10–25.34		4.26	1.86–9.76	
Pathologically renal vein invasion	−	1		0.022 *	not included		
	+	2.42	1.13–5.21				
Pathologically peri renal fat invasion	−	1		0.21			
	+	1.67	0.74–3.73				
Pathologically renal sinus fat invasion	−	1		0.22			
	+	0.612	0.27–1.34				
Pathologically invasion numbers of subgroups	1/2	1		0.0008 *	not included		
	3	14.28	3.03–67.1				

HR, hazard ratio; CI, confidence interval; CRP, C-reactive protein; INF, infiltration; LVI, lymphovascular invasion; ccRCC, clear cell renal cell carcinoma not included. We limited the number of variables three for the multivariable analysis to prevent overfitting the model because of the small number of events (26 events).

**Table 3 diagnostics-10-00154-t003:** Univariate analyses for CSS (* statistically significant).

Variables		Univariate Analysis		
		HR	95% CI	*p* Value
Gender	Male	1		0.25
	Female	2.29	0.54–9.75	
Age	≥70	1		0.81
	<70	1.17	0.33–4.08	
CRP	<0.5	1		0.28
	≥0.5	2.01	0.55–7.64	
clinically renal vein thrombus	−	1		0.049 *
	+	5.08	0.98–26.44	
Fuhrman Grade	G1,2	1		0.07
	G3,4	4.49	0.85–23.72	
INF	a	1		0.15
	b/c	3.38	0.66–17.42	
LVI	−	1		0.14
	+	2.72	0.71–10.33	
Necrosis	−	1		0.48
	+	1.56	0.45–5.42	
Histological classification	non-ccRCC	1		0.30
	ccRCC	0.42	0.10–2.15	
Sarcomatoid variant	−	1		0.5
	+	0.34	0.01–7.56	
tumor size	≤7	1		0.07
	>7	3.25	0.87–12.11	
Urinary collecting system invasion	−	1		0.07
	+	4.35	0.86–22.01	
Pathologically renal vein invasion	−	1		0.4
	+	1.69	0.48–5.86	
Pathologically peri renal fat invasion	−	1		0.21
	+	9.1	2.42–34.	
Pathologically renal sinus fat invasion	−	1		0.27
	+	1.41	0.39–5.06	
Pathologically invasion of subgroups	1/2	1		<0.0001 *
	3	NA	91–13,500	

HR, hazard ratio; CI, confidence interval; CRP, C-reactive protein; INF, infiltration; LVI, lymphovascular invasion; ccRCC, clear cell renal cell carcinoma.

**Table 4 diagnostics-10-00154-t004:** Correlation between presence of UCSI and Hematuria levels of Total/Micro/Macro.

	Spearman *r*	*p* Value
Hematuria	0.54	<0.0001
Micro Hematuria	0.43	<0.0001
Macro Hematuria	0.41	<0.0001

UCSI, urinary collecting system invasion.

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
