# Peer review of "Clinical Significance of Tumor Size, Pathological Invasion Sites Including Urinary Collecting System and Clinically Detected Renal Vein Thrombus as Predictors for Recurrence in pT3a Localized Renal Cell Carcinoma"

_diagnostics, 2020, doi:10.3390/diagnostics10030154_

Round 1
Reviewer 1 Report
diagnostics-733165
This manuscript is very impressive to present the clear impact of clinical findings (tumor size, invasion sites, and renal vein thrombosis) on the recurrence of pT3a RCC, and should be published with minor revision in discussion.
- Mechanistic insights
It would be much better to discuss on the mechanistic insights of each clinical finding with the previous (clinical and basic science) reports in order to pursue why each finding is significant for recurrence.
tumor size: as tumor is much heterogeneous and can we know the cellular composition, differentiation or proliferative status of the tumor are altered up to the size, which might affect the recurrence rate? If there are previous reports to show the single cell sequence data on this aspects, that would be great.
renal vein thrombosis: the thrombosis causing mechanisms and the status with thrombosis. Both of which
Would be important pathogenically.
- Circulating tumor cells (CTCs)
Collecting and analyzing CTCs are directly important clinically. If there are some reports of CTCs in RCC filed, it might be important to discuss on the pathophysiological importance of each clinical finding based on that.
Reviewer 2 Report
Comments to the authors: manuscript (diagnostics-733165)
General
The authors presented a straightforward study with high-risk renal cell carcinoma RCC patients. The data are conclusive and present a good overview. However, the "new" information for the clinician is small, because the clinical-pathological data of these patients are already included in the recent assessment of the patients' risk. It is therefore very welcome that the authors discuss the several serious limitations of this study. The consequence was that no factor was significantly indicative in multivariate Cox regression calculations for CSS.
Some additional points should be considered:
Major points
- The tumor size was found one of the most important prognostic factor. Tumor sizes are continuous data and already pT2a and pT2b are discriminated by the different cut-offs. It would be useful to calculate the significance of tumor size with their continuous data according to the REMARK guideline that recommended the risk calculations with continuous data, at least before a dichotomized approach is applied.
- Tables 2/3: Histological classification: "nonRCCs" generally characterized by a lower risk should be set 1, but not clear cell RCCs with higher risk
- Kaplan-Meier curves: number of patients should be indicated for the respective curve.
Reviewer 3 Report
1) Abstract, the last sentence: Is the description correct? Does early diagnosis of recurrence result in improved prognosis? The reviewer does not think so for every case. It’s better to remove the last phrase. 2) Table 1, Fuhrman grade and LVI: Please list the number of patients with missing data. 3) Table 1: Please list variables showing up in table 2 (CRP, INF etc.). 4) Table 2: Too many variables for the number of events (26 events) in a multivariable model result in overfitting of the model. Please limit the number of variables 2 or 3. 5) Table 3: The reviewer thinks multivariable analysis is not informative because of a limited number of events (11 events).Author Response
Please see the attachment

Round 2
Reviewer 3 Report
1) Table 2: The authors should not include 4 or more variables in the multivariable analysis; it would be considered as a methodological flaw and is not acceptable. The reviewer strongly recommends to include cV1, tumor size, and invasion to the collecting system, by which the authors can draw the same conclusions.
2) Table 3: The authors do not need to (should not) show a multivariable model. For the same reason as comment 1), it is sufficient to show the result of univariable analysis.
